# Unraveling Heterogeneity in Epithelial Cell Fates of the Mammary Gland and Breast Cancer

**DOI:** 10.3390/cancers11101423

**Published:** 2019-09-24

**Authors:** Alexandr Samocha, Hanna Doh, Kai Kessenbrock, Jeroen P. Roose

**Affiliations:** 1Department of Anatomy, University of California, San Francisco, CA 94143, USA; ajsamocha@gmail.com (A.S.); hanna.doh@ucsf.edu (H.D.); 2Department of Biological Chemistry, University of California, Irvine, CA 92697, USA; kai.kessenbrock@uci.edu

**Keywords:** mammary gland, breast cancer, cell fate, heterogeneity, 3D cultures, organoids, signaling, single-cell RNAseq

## Abstract

Fluidity in cell fate or heterogeneity in cell identity is an interesting cell biological phenomenon, which at the same time poses a significant obstacle for cancer therapy. The mammary gland seems a relatively straightforward organ with stromal cells and basal- and luminal- epithelial cell types. In reality, the epithelial cell fates are much more complex and heterogeneous, which is the topic of this review. Part of the complexity comes from the dynamic nature of this organ: the primitive epithelial tree undergoes extensively remodeling and expansion during puberty, pregnancy, and lactation and, unlike most other organs, the bulk of mammary gland development occurs late, during puberty. An active cell biological debate has focused on lineage commitment to basal- and luminal- epithelial cell fates by epithelial progenitor and stem cells; processes that are also relevant to cancer biology. In this review, we discuss the current understanding of heterogeneity in mammary gland and recent insights obtained through lineage tracing, signaling assays, and organoid cultures. Lastly, we relate these insights to cancer and ongoing efforts to resolve heterogeneity in breast cancer with single-cell RNAseq approaches.

## 1. Introduction into Mammary Gland Structure, Function and Early Development

The mammary gland is comprised of many different interacting cell types. Epithelial cells form a primitive structure early during embryonic development, which later form the nipple and a ductal network that expands into the fat pad. These epithelial ducts are surrounded by fat cells, and become innervated with a variety of stromal cells including endothelial cells, immune cells, and fibroblasts [1]. Maturation of the gland is regulated at first by mesenchymal interactions, and later by hormone and growth factor receptor signaling during puberty and pregnancy. Mammary epithelial cells assemble into their normal morphology between E16-E18 (embryonic day), forming a bilayered duct with an inner lumen [2]. Together, the mammary epithelial subtypes interact to carry out the organ’s functions. After pregnancy, mature epithelial cells can later differentiate into alveolar cells and subsequently produce milk proteins.

Mammary epithelial cells (MECs) typically form a bilayered epithelium and can be broadly separated into two distinct compartments: an inner layer of luminal and outer layer of basal MECs. However, additional heterogeneity exists within both luminal and basal MECs. Luminal populations are often sub classified based on hormone and growth factor receptor status. The basal epithelium contains a subset of so-called myoepithelial cells that lie along the outside of the ductal epithelial tree and assist in the motility of milk protein along the lumen. The MEC system also contains subsets of basal and luminal stem and progenitor cells, which will be covered in greater detail later. For example, within the luminal compartment a population of luminal progenitor cells can go on to form ductal or alveolar cells. In addition, basal cells contain a subset of mammary stem cells (MaSCs) forming a small, heterogeneous, and to date poorly defined population that drive development, repair and organ reconstitution when transplanted into epithelium-cleared fat pads of recipient mice [3].

Puberty is the most dynamic and striking period during the development of the mammary gland. The rudimentary duct undergoes significant expansion, resulting in the formation of bulbous multilayered structures called terminal end buds (TEBs) [4] (Figure 1). TEBs are the proliferative centers that drive elongation, bifurcation, and branching until the entirety of the mammary fat pad is filled, thereby creating the mature epithelial tree [5]. The TEB contains cap cells, a rapidly growing and dividing progenitor cell population that later goes on to form the tree’s outer myoepithelial layer [6].

## 2. Luminal Cells and Luminal-Specific Progenitors

The luminal compartment contains different mature, differentiated cell populations with specified functions as well as progenitor-cell populations that are each distinguished by specific gene and extracellular ligand profiles.

Several classifications of luminal cells exist within the mammary gland. Luminal cells are segregated based on functional, morphological, and expression profiling evidence. Histologically, luminal cells can be broadly separated into two types: ductal and alveolar. Mature ductal cells can be either estrogen receptor (ER) positive or negative, and feature the following marker expression profile: CD49f^lo^CD29^lo^CD24^+^CD14^−^EpCAM^hi^c-kit^−^Sca1^+^CD61^−^CD49b^−^ [7], in which the CD numbers and other markers represent different gene products. Mature alveolar cells are always estrogen receptor (ER) negative and can be more precisely characterized through a CD49f^lo^CD29^lo^CD24^+^CD14^−^EpCAM^hi^Sca1^lo^CD61^−^ expression pattern. Alveolar cells can be distinguished by their secretory morphology and histologically by the accumulation of milk proteins within.

Transcription factors have been demonstrated to drive luminal cell fate commitment throughout development. The Elf5 (E74-like factor 5) transcription factor is one of the major drivers of luminal cell specification and differentiation. Elf5 is able to directly repress Slug (Zinc Finger Protein SNAI2) transcription, thereby blocking basal cell determination and promoting luminal cell fate [8,9]. Progesterone acts on mature, progesterone receptor-positive cells and induces expression and secretion of the chemokine RANKL (Receptor activator of nuclear factor kappa-Β ligand) that subsequently, in a paracrine manner, induces progenitor cells to express Elf5 [10]. The transcription factor Gata-3 (GATA binding protein 3) is another key regulator of luminal cell identity. Gata-3 expression is important for the maturation of luminal progenitor cells into mature ductal and alveolar cells [11]. Stat5a (Signal Transducer and Activator of Transcription 5A), a transcription factor activated by activated by ligands including prolactin, growth hormone, and EGF (Epidermal Growth Factor) drives the expansion of luminal progenitors and subsequent differentiation into alveolar cells [12].

Luminal progenitor populations can be distinguished using many of the above extracellular markers. For example, ductal progenitors are CD61^+^, CD49b^+^, CD14^+^, and c-kit^+^ whereas their mature counterparts are negative for the surface expression of these proteins [13,14,15]. Sca-1 (Spinocerebellar ataxia type 1) expression can be used to separate ductal progenitors from alveolar progenitors since alveolar progenitor cells lack membrane expression of the protein. It should be noted that expression of these extracellular receptors is not ubiquitous between mouse model lines. Neither c-kit nor CD61 are expressed in mammary epithelial cells derived from C57BL/6 mice, while they are both found in MECs from FVB/N mice [16]. The fate and biology of these luminal progenitor cell populations has been further characterized by lineage tracing experiments, which we will discuss in Section 4.

## 3. Basal Cells and Progenitors

Unlike luminal cells, the factors that control and label different basal cell populations are less clear. Both stem cells and bipotent progenitor cells have been identified within the broader basal cell lineage. The use of extracellular markers to distinguish these discrete populations, however, has not yet yielded clear profiles. Mature myoepithelial basal cells are typically CD29^hi^CD49f^hi^CD24^+^EpCAM^lo/med^ [7]. The undifferentiated population that is enriched for MaSCs is believed to be CD49f^hi^CD29^hi^CD24^hi^EpCAM^lo/med^Sca1^−^. Markers that are specific for a restricted basal cell progenitor have not been identified and neither have specific markers for a restricted myoepithelial cell progenitors. Luminal progenitors are thought to arise from an early basal cell population.

Several epithelial-mesenchymal transition (EMT) factors have been identified as molecular regulators of the basal/MaSC population. The transcription factor Slug is a key for the basal/MaSC lineage as it suppresses luminal cell fate. Slug-deficient mice have delayed ductal morphogenesis and aberrantly express luminal signature genes in basal cell populations [17,18]. The transcription factor Sox9 cooperates with Slug to suppress luminal fate and exogenous expression of these two factors can convert differentiated luminal cells into MaSCs with long-term mammary gland-reconstituting ability [18]. Tumor protein TP63 loss blocks the formation of the primitive epithelial duct [19]. Perhaps one of the strongest drivers of basal cell identity, p63 overexpression in luminal cells leads to an identity shift from luminal to a basal phenotype [20]. Notch1/3 signaling has been reported to down-regulate TP63 expression as basal progenitors restrict to a luminal cell fate [21]. The transcription factor p53, on the other hand, restricts basal/MaSC renewal and drives differentiation [22].

## 4. Lineage Tracing to Identify Restricted and Bipotent Progenitor Cells

Lineage tracing is a powerful tool to observe and track the function of adult-, stem- and progenitor-cells during normal homeostasis, and various studies have utilized this approach to shed light on the lineage hierarchies of the MECs. K8-creER targeted cells exclusively differentiate into luminal cells when observed in vitro [23]. MECs marked with this label are maintained in small numbers following several rounds of growth and differentiation. Elf5, a well-studied luminal progenitor cell gene, was utilized in a separate study to trace this restricted progenitor population throughout pubertal development [24]. Elf5-expressing cells exclusively produced mammary epithelial cells with a luminal identity and were a major driver of branching morphogenesis [24]. In 2014, Rios et al. confirmed that these Elf5-expressing cells were short-lived and restricted progenitors, as the positive cells could not be detected in the mammary glands of lineage tracer mice after 20 weeks [24]. Combined, these findings show that there may exist both short- and long-lived progenitor cells that exclusively mature into luminal subtypes.

Lineage tracing studies were performed in order to hone-in on the exact nature of mammary progenitor populations. In 2011, van Keymeulen et al. established that all mammary epithelial lineages are derived from cytokeratin-14 (K14) expressing embryonic progenitors [23]. Following this population throughout puberty revealed that K14+ MaSCs promote basal expansion and maintenance. MaSCs expressing K8 promoted luminal cell expansion and maintenance throughout puberty. When cultured in vitro these restricted progenitors maintain their identity, while only basal cells display multipotency in transplantation and reconstitution assays [23]. Rios et al. used lineage-tracing experiments to identify a cytokeratin 5 (K5) expressing bipotent progenitor population [24]. K5-positive cells labeled at the onset of puberty contributed to the formation of both mature luminal and myoepithelial cells.

## 5. Three-Dimensional Spheroid Cultures

In the late 1980s and early ′90s, three-dimensional (3D) cultures were instrumental in the discovery of the cell biology of mammary gland development and breast cancer. We term these classical three-dimensional spheroid cultures in our review, as these did not yet focus so much on the function and preservation of stem cells through Wnt signaling. We will cover more on Wnt signals and organoids later. For an excellent overview with historic perspectives on the development of these 3D culture platforms we refer you to a review by Drs Simian and Bissell [25]. Essential for these revolutionary in vitro approaches was the isolation of gel from the matrix of chondrosarcomas in 1977 by Orkin and colleagues [26] that is now commonly known as Matrigel (Figure 2). For example, these classical 3D spheroid cultures revealed that a 3D basement membrane [27] impacts the response of a series of human breast-tumor cell lines at different stages of progression, cultured within a physiological context [28]. Importantly, the 3D aspect of these cultures revealed new discoveries, such as bidirectional cross-modulation of integrin and EGFR (Epidermal Growth Factor Receptor) signaling, that were not present in artificial 2D cultures [29].

Classical 3D spheroid cultures also aided identification of restricted luminal- or basal-cells and bipotent progenitors. In vitro experimentation confirmed that, the restricted differentiation potential of both luminal cells and basal, myoepithelial cells can be maintained, given specific culture conditions [30]. In mammosphere assays, single MECs can be cultured short-term in ultra-low adherence plates to generate mammospheres. These assays interrogate the function and clonogenic capacity of MaSCs. Mammosphere cultures could generate MEC colonies with either restricted unipotent differentiation potential or a bipotent stem-like phenotype, suggesting the existence of unipotent- and bipotent-progenitors [31]. Isolated single mammary epithelial cells can also be plated in Matrigel pellets in a colony-forming assay (Figure 2). Single MECs are able to seed into the surrounding matrix and clonally expand. The number of colonies that are able to form, as well as the size of the colonies, provides a representation of the proliferative capacity of progenitor cells within the total MEC population. Similarly, only MaSCs with self-renewal potential are able to survive non-adherent conditions. The number of mammospheres that are able to form can be used as a proxy to quantify MaSC activity. Use of these platforms has allowed for new insights into the mechanisms by which ductal morphogenesis is directed, particularly with regard to the role of different growth factor receptors, which we discuss next.

## 6. EGFR Family and Ligands

The EGFR family consists of EGFR (ErbB1/Her1), ErbB2/Her2, ErbB3/Her3, and ErbB4/Her4. ErbB receptors can both homo- and heterodimerize, with 10 possible combinations-six heterodimers and four homodimers. There is a high level of homology between kinase domains of the four EGFRs (~60–80%) with divergence occurring predominantly in the C terminus (shared identity is only ~10–25%). ErbB proteins have tissue specific expression, though they are commonly expressed together. EGFRs are found in several epithelial cell types such as the lung, intestine, and the breast. ErbB2 has no known ligands (Table 1), but frequently dimerizes with the other three EGFR members because of its unique and extended interaction loop [32]. ErbB3 is kinase dead, its cytoplasmic domain unable to initiate phosphorylation cascades. ErbB3 is able to trans phosphorylate its own intracellular domain to assist heterodimerization, as well as allosterically activate other EGFRs, however [33].

Up to 13 different ligands have been found to bind EGFR family proteins: EGF, HB-EGF, transforming growth factor (TGF), amphiregulin (AREG), epiregulin (EREG), epigen (EPG), betacellulin (BTC), and neuregulins 1–6 (NRG) [34,35,36]. Ligand binding induces phosphorylation at tyrosine residues on the cytoplasmic domain [37]. EGF and TGF are the key ligands for EGFR and ErbB3/B4 preferentially bind neuregulins. The EGFR family members and their preferred ligands can be found in Table 1. Evidence suggests that EGFR phosphorylation and the duration and amplitude of signaling events are influenced by the binding of different ligands. This results in divergent cellular responses. For example, AREG is more potent in stimulating ductal elongation compared to EGF [38]. These ligand-specific nuances are important during development and cancer and we will cover some of the cancer-specific nuances of EGFR signaling in Section 10.

## 7. EGFR during Mammary Gland Development

The vast majority of what is understood about the EGFR family’s role in mammary development comes through the use of genetic mouse models. *Egfr*-deficient mice perish just after birth, which complicated deciphering the exact nature of EGFR during mammary organogenesis [39,40]. Luetteke et al. generated *Waved-2* mice to circumvent this problem. The *Waved-2* allele has a point mutation near EGFR’s cytoplasmic kinase domain that reduces activity; it is hypomorphic [41]. *Waved-2* mice have defective mammary development with diminished branching and a reduction in ductal invasion [42,43]. Use of a dominant negative EGFR protein using the mammary-specific MMTV promoter confirmed its role during pubertal development. Mice with the dominant negative EGFR display reduced proliferation and inhibited duct maturation [44]. *Egfr*’s importance in the stroma was confirmed via the generation of mixed tissue recombinants from transplanting neonatal epithelial cells from wild type or *egfr*-deficient mice [43,45,46].

Perturbations in other EGFR family genes also result in dramatic mammary developmental phenotypes. Specifically, deficiency in the *ErbB2–4* results in impaired ductal outgrowth during puberty. Deletion of *ErbB2* shunts ductal outgrowth [47,48]. ErbB2 also controls terminal end bud (TEB) formation through its regulation of cellular compartmentalization. In summary, despite many studies into the role of EGFR proteins in the mammary gland, the exact nature of each member has not been fully elucidated. Stromal and epithelial expression of the EGFR family is critically important at all stages of mammary development. A better understanding of EGFR and its downstream effectors is needed to create a clearer picture of the signals and processes that regulate the complex process of mammary organogenesis.

## 8. EGFR Signal Strength, Downstream Effector Kinases, Cell Fate

Mammary epithelial cells are organized into a developmental hierarchy based on extracellular receptor and gene expression patterns. The exact nature of these populations, and the factors that balance their proliferation with differentiation, are not well understood. Recent evidence has emerged, however, that EGFR signaling in MECs may be a key player in better defining this hierarchy as depicted in Figure 3.

In 2011, a report by Pasic et al. began to decipher EGFR’s potential role in controlling MEC fate decisions during development. An ex vivo organoid model was utilized using cells taken from normal human breast tissue. They observed that different EGFR ligands could elicit discrete cell fate decisions. EGF stimulation of human breast organoids initiated a significant expansion of the basal (myoepithelial) population. Conversely, AREG stimulation drove organoids towards a luminal (ductal) cell fate. Interrogation of the downstream effector revealed that this deviance in cell fate decisions was due in part to the strength of downstream MEK-ERK signals, in which stronger activation EGFR-Ras-MEK-ERK selectively expanded the basal cell population and weaker activation drives luminal expansion [49].

Mukhopadhyay et al. expanded our insights into this initial model in 2013 [50]. Using an hTERT-immortalized human stem/progenitor cell pool, they observed similar cell fate decision changes that were dependent on the strength and duration of EGFR signals. Once more, it was observed that stimulation with the weak agonist AREG promoted luminal cell fate and a strong agonist (TGFα) drove cells towards a basal cell identity. In contrast to the data presented in Pasic et al. [49], however, Mukhopadhyay et al. found that EGF stimulation did not drive MaSCs down a specific lineage [50]. The addition of U0126, an inhibitor against the MEK-ERK pathway, significantly reduced differentiation into CD49f^lo^EpCAM^hi^ and EpCAM^lo^ cells [50]. Taken together, it appears that the duration and amplitude of EGFR signals affects MEC fate choices.

Since many of the signaling effectors triggered by the EGFR lay downstream of Ras, it is of interest to consider the strength and duration of Ras activation as the cell fate determination factor. A historic study reported that nuances in receptor-Ras signaling can affect cell fate in a PC-12 cell line system. Stimulation of rat adrenal carcinoma cells (PC-12) with different EGFR ligands produced altered cell fate. In the PC-12 system, EGF is a weaker agonist compared to the strong nerve growth factor (NGF). EGF stimulation led to a short pulse of Ras-MEK-ERK activation and cell proliferation, while NGF stimulation elicited prolonged Ras-MEK-ERK signals, exit from the cell cycle and differentiation [51]. Since that report in 1995, very little work has followed up on this, perhaps because it is challenging to couple quantitative biochemical measurements to cell fate decisions, especially in in vivo studies.

## 9. EGFR and Other Receptor Tyrosine Kinases Signaling and Ductal Morphogenesis

Ductal morphogenesis is the process in which the mammary epithelium invades the fat pad during puberty to form a fully branched ductal epithelial tree. It is known to occur in a somewhat stochastic process, regulated by the combinatorial input of diverse signals. The stochastic aspect is perhaps best exemplified by the fact that there are substantial differences between mammary glands of mouse littermates. This has led the field to conclude that predetermined genetic control of pubertal development is not a possibility, unlike in the development of other epithelial tissues [52]. Development is dictated by mechanical factors and molecular signals from the surrounding stroma [53]. As a result, maturation of the breast is context-dependent.

The pubertal developmental process is initiated in large part by the expression of ovarian and pituitary hormones [54]. These signals cooperate to facilitate growth and communication between epithelial and stromal cells. Genetic knockout of estrogen receptor alpha (ERα) leads to incompletely developed mammary ductal trees, and exogenous administration of estrogen in mice lacking ovaries rescues ductal morphogenesis [55,56]. Estrogen facilitates stromal cell expression of hepatocyte growth factor (HGF) and epithelial expression of amphiregulin (AREG) [57,58]. AREG can communicate with the epidermal growth factor receptor (EGFR), whose expression is essential on mammary stromal cells [46]. Estrogen’s collective functions serve to regulate local cell growth during pubertal development.

Estrogen is not the only steroid hormone important for pubertal mammary gland development, however. In fact, estrogen alone is insufficient to restore ductal morphogenesis when other input (like from the pituitary gland) is missing. Exogenous administration of growth hormone, normally produced by the pituitary gland, can restore the impaired branching phenotype of pituitary gland-deficient mice [59]. Growth hormone induces stromal insulin-like growth factor 1 (IGF1) expression, which binds to IGFR on epithelial cells [60]. Together growth factor and IGF1 act as global regulators of ductal morphogenesis.

Several growth factor receptors and receptor tyrosine kinases (RTKs) are involved in the integrated signaling environment that directs mammary morphogenesis. Fibroblast growth factors (FGF) and their receptors (FGFRs) are critically important for growth and branching. FGF2 and FGFR2 in particular shape the profile of ductal outgrowth and MEC proliferation. Genetic deletion of *fgfr2* disrupts ectodermal and placode formation during embryonic mammary organogenesis [61]. Mammary epithelial cells lacking *fgfr2* have a proliferative disadvantage when compared to their wild-type and *fgfr2*-heterozygous counterparts, and are depleted within TEBs [62]. Mice with inducible deletion of *fgfr2*^−/−^ reveal a similar phenotype; proliferation is significantly attenuated and TEBs are completely absent from the glands [63]. In summary, *fgfr2* is a key regulator of luminal epithelial cells and it plays a specific role in the TEBs of elongating ducts.

The EGFR family of signaling proteins plays a pivotal role in directing pubertal mammary gland development in conjunction with FGF and FGF2. In brief, perturbations to each of the four EGFR family members result in developmental defects. In ex vivo culture, FGF2 addition can rescue growth and branching in EGFR-null 3D spheroids [45]. Mice with dominant negative EGFR display reduced proliferation and inhibited duct maturation [44]. Deletion of *ErbB2* shunts ductal outgrowth [47,48]. *ErbB2* also controls TEB formation through its regulation of cellular compartmentalization. *ErbB3* deficiency results in small TEBs and increased branch density with decreased TEB size as a result in an increase in apoptosis, controlled via observed changes in the PI3K (Phosphoinositide 3-kinase) signaling pathway. Anti-apoptotic transcription factor Bcl-2 is reduced in *ErbB3^−/−^* mice [64]. Mice deficient for *ErbB4* have defect occurring later in breast development, specifically during the formation of milk-producing luminal cells [65].

## 10. ERFR in Breast Cancer and Oncogenic PI3K Signals That Switch Fate

Dr Schlessinger put forth an elegant and simple model for EGFR activation with EGF as an external signal leading to the conversion of a monomeric receptor to a ligand-induced dimer [66], which served as a framework for further studies. The EGFR dimer turned out to be an asymmetric structure in which one EGFR kinase domain in a dimer acts as an allosteric activator for the other [67], which subsequently paved the way to mechanistically understand how a catalytically inactive Her3 can facilitate the activation of other EGFRs, such as Her2 [68]. Her2 overexpression is frequently found in breast cancer and typically associated with poor prognosis; a Her2/Her3 heterodimer operates as an oncogenic unit that drives breast cancer proliferation [69], with the phosphorylated tyrosines in the intracellular tail of the catalytically inactive Her3 functioning as adapter scaffolds for intracellular kinases such as PI3K (Phosphoinositide 3-kinase) and PLC (Phospholipase C).

Whereas it is clear that the EGFR family plays a role in mammary gland development and maturation as well as breast cancer, a lot less is known about the specific effector kinase pathways that lie downstream of the receptor families such as the EGFR. Elegant imaging studies revealed that levels of phosphorylated ERK kinases (a type of MAPK) are highest in MECs near the front of elongating ducts whereas levels of phosphorylated Akt kinase were equal throughout and Huebner et al. proposed a model of receptor-induced proliferation leading to highly motile cells with high phospho-ERK [70]. Two thought-provoking studies relate cell fate choices to aberrant PI3K signals in the context of breast cancer. Cell fate, heterogeneity, and cell lineage conversions are aspects that are of great relevance to cancer as these impact the tumor type and also often responses to therapy [71]. PIK3CA (Phosphoinositide 3-kinase p110 alpha) with a histidine to arginine mutation at position 1047 (H1047R) is a frequent mutation occurring in human breast cancer and expression of PIK3CA(H1047R) in lineage-committed basal Lgr5-positive and luminal keratin-8-positive cells led to dedifferentiation into a multipotent stem-like state [72]. Furthermore, the tumor cell of origin influenced the frequency of malignant breast tumors, linking (heterogeneous) cell fate to cancer aggressiveness [72]. Van Keymeulen et al. reported similar results in their studies in the context of loss of the tumor suppressor p53; expression of PIK3CA(H1047R) in basal cells (keratin K5-CreERT2 driver) gave rise to luminal-like cells and expression in luminal cells (K8-CReERT2) resulted in basal-like cells before progressing into invasive tumors [73]. Therefore, the rules that are thought to exist in normal unipotent progenitors (see Figure 3) appear alter when cells express PIK3CA(H1047R), which brings up a bigger question. What aberrant biochemical signals are capable to induce crossover of basal- and luminal-cell fates and drive further increases in heterogeneity and fluidity between mammary epithelial cells?

## 11. The Mammary Stem Cell Conundrum: More Questions than Answers

While there is a general consensus about some of the factors that mark and or regulate the MaSC population, the field has been left with more questions than answers when trying to specifically define MaSCs [74]. For example, the estimates for stem cell frequency, derived from calculating the number of mammary repopulating units, varies dramatically. Estimates for MaSC frequency range from 1 in every 100 total cells to 1 in every 4900 [75,76,77]. Extracellular markers used to sort MaSCs have proven insufficient to exclusively identify MaSC populations, but rather, only identify populations within which MaSCs are enriched [78,79,80]. The embryonic stem cell marker Oct4 (Octamer-binding transcription factor 4) has been found to label human mammary stem cells, but is insufficient to specifically separate MaSCs from other progenitor cells [81,82].

Some progress has been made towards separating MaSCs from mature myoepithelial and basal progenitors by using transgenic mouse models. MaSCs residing in the terminal end bud (TEB) cap cells exclusively express the phosphatase s-Ship [83], but MaSCs during pubertal development are not just found in cap cell populations. Lgr5, given its well-known role in intestinal epithelial populations, became of interest as a potential marker of the MaSC population [84]. Again, the Lgr5 story is unclear with opposing results of studies that use slightly different approaches. Whether Lgr5^+^ cells have increased repopulating activity and whether they trend towards quiescence or self-renewal has not been clearly delineated. The exact nature and potential of Lgr5^+^ and Lgr5-negative MaSC populations is an ongoing field of research [85,86,87]. The Lgr5 question is emblematic of the larger picture that MaSCs are a highly heterogeneous population. Isolating a basal population enriched for MaSCs from fetal mice revealed a distinct stem cell identity. Fetal-derived MaSCs display augmented clonogenic potency and repopulating efficiency when compared to their adult stem cell counterparts [88]. Further, these fetal MaSCs expressed unique extracellular receptor pattern, one that has features of both basal and luminal cells, which was also different from pubertal MaSCs [89]. The heterogeneity is further compounded when looking at adult mammary glands. Adults MaSCs have been found to display “lineage priming,” in which restricted differentiation programs exist within seemingly pluripotent and self-renewal competent stem cells [90,91]. Recent RNAseq studies have begun to further subdivide the MaSC pool. In 2017, Pal et al. found an early progenitor subset marked by CD55 that exists between basal and luminal cells [92]. In sum, the spatial-, temporal-, and functional-heterogeneity of MaSCs has made their study complicated and bolstered the importance of finding new regulatory factors.

In summary, in vitro and in vivo findings suggest that breast epithelial cells can be arranged into a developmental hierarchy (Figure 3), but it should be explicitly stressed that this is just a model. In this hierarchy, mature luminal and myoepithelial cells are maintained by lineage-restricted progenitors, which is replenished by the MaSC population [93]. Bipotent progenitors exist within this hierarchy, but the exact placement is unclear and has therefore been excluded. Extracellular ligand expression and transcription factor regulation has served as the basis for the construction of this model. The extraordinary complexity of the mammary stem cell population has complicated a clear understanding of the drivers for epithelial cell fate decisions.

## 12. Organoids to Assess Stem- and Progenitor-Potential

Organoids are miniature, three-dimensional, in vitro tissue cultures that retain stem cell function, and are generated from pieces of tissue (Figure 4). Organoids have been instrumental to study stem cell biology of epithelial cell lineages [94,95]. They have gained traction as ideal platforms to screen for biomarkers, obtain personalized predictive/prognostic information, and test novel therapeutic strategies and rational drug design [96,97,98]. Organoids and classical mammosphere cultures are similar in that both are in vitro 3D cultures of cells (Figure 2 and Figure 4). However, organoids typically depend more on the extracellular matrix and may display more self-organization and spatially restricted lineage commitment [99].

Furthermore, organoids can be cultured indefinitely and can be cryopreserved, whereas classical mammosphere cultures generally survive for shorter time periods [100]. Indeed, several different mammosphere cultures media cocktails have been applied in the past, but for many of these studies the goal was not to sustain growth long term. For example, Mroue and Bissell used growth media containing insulin transferrin selenium (ITS) and fetal bovine serum (FBS) for mammospheres generated from mouse mammary glands, but cultures were not sustained for long periods of time in this study [101]. It should be noted that the use of FBS in organoid culture is challenging since the exact composition is unknown and components can vary from batch-to-batch. Ewald et al., used growth media containing ITS supplemented with FGF2 and cultures were grown and assessed for a couple of days [102].

A crucial part of successfully culturing organoids (Figure 4) is to preserve stem cell function over long periods of time. The Wnt pathway has been shown to play an important role in stem cell maintenance [103]. In the canonical Wnt pathway, Wnt proteins bind to Frizzled (FZD) and LRP (lipoprotein receptor-related protein) receptors. Dickkopf (DKK) is a ligand that binds to LRP6 with high affinity [104]. Wnt binding to FZD and LRP receptors results in increased levels of β-catenin, which translocate into the nucleus to form a transcriptionally active complex with Tcf factors (T cell factor) [105,106]. In the absence of Wnt signals, Tcf transcription factors bind to Groucho repressors [107,108,109] and as such Tcf factors act as switches to turn on Wnt target genes [110]. Intriguingly, one of the β-catenin/Tcf target genes is the extracellular receptor *Lgr5* [111]. R-spondins are ligands for Lgr5 (Leucine-rich repeat-containing G-protein coupled receptor 5), and can associate with the Frizzled/LRP Wnt receptor complex to enhance the Wnt signal [112]. Structure-based design and subsequent production of surrogate Wnt ligands that are hybrid molecules combining the DKK and Wnt ligands proved clever tools to sustain canonical Wnt signaling in combination with R-spondin ligands and support efficient organoid growth [113].

In addition to surrogate Wnt and R-spondin ligands, EGF and Noggin are required to indefinitely expand organoids [114]. Furthermore, additional components can be added to the organoid medium to help maintain organoid cultures. These include fibroblast growth factor (FGF) 7, FGF10, Activin like kinase inhibitor (A83-01), SB202190 (p38 mitogen-activated protein kinase inhibitor), and nicotinamide [115]. Neuregulin 1 (Nrg1) is required for morphogenesis and differentiation of the mammary gland [116]. Addition of Nrg1 to organoid culture medium resulted in higher efficiency of mammary organoid generation [114]. Rho kinase inhibitor (Y-27632) was found to induce indefinite proliferation in vitro in normal and tumor epithelial cells [117] and addition of Y-27632 to organoid culture medium improved organoid culture conditions [114]. In Section 6 we discussed the heterogeneity and complexity of the stem- and progenitor- cell population in the mammary gland with Lgr5+ and Lgr5− cells and different characteristics in fetal-, puberty-, and adult-stages. We anticipate that successful mammary gland organoids will require complex but clearly defined (growth) factor medium and that systematic application of standard operating protocols (SOPs) will propel the field’s efforts of organoid and cell heterogeneity forward.

## 13. Reconstructing Mammary Epithelial Cell Types and States Using Single-Cell Genomics

Advances in next generation sequencing and microfluidic-based handling of cells and reagents now enable us to explore cellular heterogeneity in an unbiased manner using various single-cell genomics modalities to profile genomic features in individual cells [118]. The current scientific knowledge about the MEC system is largely limited to data generated by bulk profiling methods, which only provide averaged read-outs that generally mask cellular heterogeneity. This averaged approach is particularly problematic when the biological effect of interest is limited to only a subpopulation of cells such as stem/progenitor cells, which may comprise only minor subsets of the total number of cells in a tissue. However, over the very recent years several studies emerged that utilized single-cell genomics approaches for unbiased identification of the cell types and states within the MEC compartment.

Among the genomics modalities available to date, single-cell RNA sequencing (scRNAseq) is the most advanced and most widely accessible to the research community, and has recently been applied to both human and mouse mammary epithelial samples (Figure 5). A recent study using a combination of microfluidic- and droplet-enabled single-cell transcriptomics pipelines revealed that the human breast epithelium contains three very distinct types of cells that each contain additional distinct cell states [119]. Based on these data, the human epithelium contains one basal and two luminal cell types, namely a hormone-responsive and a secretory type of luminal cells. The secretory luminal cell type was previously called luminal progenitors and generally expresses markers such as ELF5 (E74-like Facotr 5) and KIT (v-kit Hardy-Zuckerman 4 feline sarcoma viral oncogene homolog, CD117). Within basal cells, further distinctions can be made based on expression of myoepithelial markers (e.g., ACTA2, Actin Alpha 2, Smooth Muscle). Pseudo temporal analysis of the differentiation trajectories in this dataset defined a continuous lineage that seamlessly connected the basal and two luminal cell types, which is in line with the concept that basal and luminal cell types are maintained by an integrated system of stem and progenitor cells.

The mouse mammary epithelium at post-pubertal stage generally revealed a comparable cell type and state composition with some differences in terms of marker genes expressed in each cell type. Pal et al. [92] used scRNAseq of isolated mouse MECs to define three main cell types, namely basal (marked by Krt14, Keratin 14), secretory luminal or also called luminal progenitors (L-sec; marked by *Elf5*) and mature, hormone-responsive luminal cells (L-HR; marked by Prlr, Prolactin receptor). In addition to the main cell types, intermediate states were identified marked by expression of both luminal and basal genes, which could be indicative of transitional cell states between luminal and basal cells and thus support the notion of one continuous lineage trajectory that maintains all MEC cell types. Another similar study focused on the differentiation dynamics of the mouse mammary epithelium during various developmental stages of adult virgin, pregnant, lactating and involuting mammary gland using scRNAseq, which defined the lineage hierarchies as a differentiation continuum rather that discrete differentiation stages [120]. The most elusive cell type remains the MaSC, which so far did not emerge as a distinct cluster in either human or mouse single-cell transcriptomics analyses, although a subset of basal cells in both human and mouse cells expresses the putative MaSC markers PROCR (Protein Coding, CD201) or LGR5 [119,120]. It remains to be determined whether more sophisticated analysis tools such as single-cell potency analysis [121], or higher sequencing depth per cell and larger cell numbers are required to unambiguously define the MaSC as a distinct cell type (Figure 5). Recent years have also seen the emergence of studies that applied scRNAseq to unravel complexity of breast cancer [122,123], including the immune environment of breast tumors [124,125], and response to Herceptin therapy that targets EGFR2 (Her2) [126]. More is certainly to come as scRNAseq and its analysis becomes more mainstream and affordable.

In addition to gene expression programs, the epigenetic makeup of the cell is a critical determinant of cellular identity that is not detectable in scRNAseq data. Recent technological advances now allow for profiling chromatin accessibility using the Assay for Transposase-Accessible Chromatin using sequencing (ATACseq) to reconstruct cis/trans regulatory elements associated with cellular identity [127]. Further adaptation of this pipeline enabled single-cell-level ATACseq to profile cellular heterogeneity on an epigenetic level both using massively parallel [128] and combinatorial indexing methods [129]. This approach has recently been applied to elucidate transcriptional regulators of the fetal mammary gland developmental lineages showing that fetal MaSCs can be separated into basal-like and luminal-like lineages, suggesting an early lineage segregation prior to birth [130]. Another study utilized a combination of single-cell ATAC and RNA sequencing isolated mammary epithelial cells to reveal the spectrum of heterogeneity within the MEC system in the adult stage [131] Interestingly, a distinct luminal progenitor cell state within the secretory luminal cell type emerged in chromatin accessibility analysis that was clustering separately in transcriptomics data. By integrating single-cell transcriptomics and chromatin accessibility landscapes, this work further identified novel cis- and trans-regulatory elements that are differentially activated in the epithelial cell types and the newly defined progenitor cell state. Taken together, these single-cell genomics studies provide invaluable resources that may serve as reference atlases to map out how the system goes awry during diseases such as cancer in unprecedented resolution.

## 14. Conclusions

In summary, cartoons such as the one depicted in Figure 3 in our review give the impression of simple and clean cell lineage choices with well-defined trajectories of cell development in unipotent manners. In reality the mammary gland, and therefore breast cancer, is much more complex and cellular heterogeneity is obvious. The pair of studies using expression of oncogenic PIK3CA(H1047R) in different mammary epithelium cell subsets give particularly striking examples of crossover between basal- and luminal- cell fates and the possibility of dedifferentiation into a multipotent stem-like state [71,72]. It is not difficult to imagine how such fluidity in cell identity may greatly impact how breast cancer patients respond to specific types of therapy in the clinic. Fortunately, technology is constantly evolving. For example, studies that assess the potential of cell populations with organoids coupled to characterization of the transcriptional landscape at the single-cell level are starting to emerge (Figure 6). Intelligent combination of organoids, mouse models, scRNAseq, ATACseq, lineage tracing, CyTOF (mass cytometry by time-of-flight), and other novel technology platforms will be required to comprehensively understand cell heterogeneity in the mammary gland and breast cancer.

## Figures and Tables

**Figure 1 cancers-11-01423-f001:**
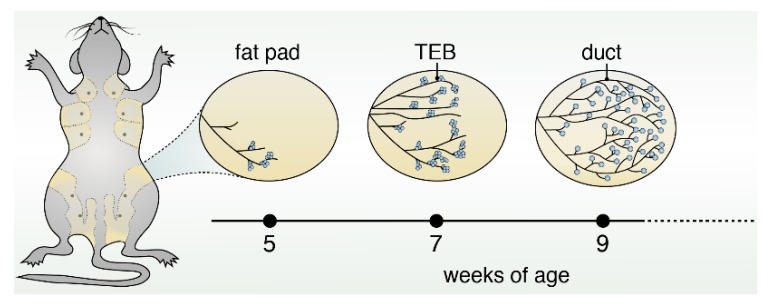
Schematic of mouse mammary gland development during puberty. During puberty, the rudimentary duct undergoes significant expansion, resulting in the formation of bulbous multilayered structures called terminal end buds (TEBs). These TEBs are the proliferative centers that drive elongation, bifurcation, and branching of ducts until the entirety of the mammary fat pad is filled, thereby creating the mature epithelial tree.

**Figure 2 cancers-11-01423-f002:**
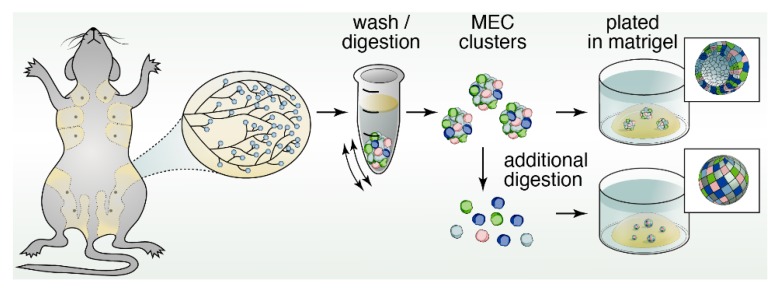
Schematic of three-dimensional spheroid cultures of mammary gland tissue. The discovery of Matrigel by Orkin and colleagues [26] enabled cell biological studies by many research groups with three-dimensional (3D) spheroid cultures of mammary gland tissue in vitro. Plating of digested MEC clusters aided the identification of restricted luminal- or basal-cells and bipotent progenitors in 3D, while further digestion into single mammary epithelial cells and plating into colony-forming assays provided information on the proliferative capacity.

**Figure 3 cancers-11-01423-f003:**
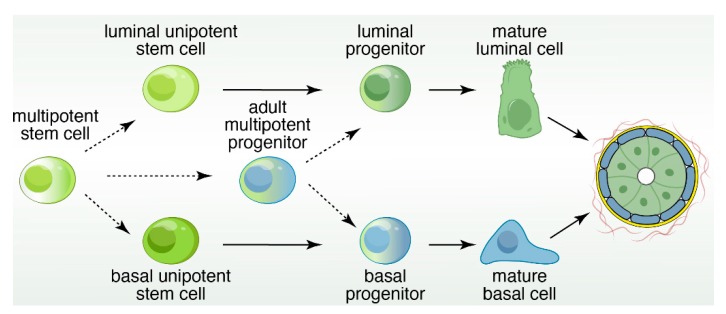
Schematic of the developmental hierarchy in the mammary gland. It should be explicitly stressed that Figure 3 is a model. In this hierarchy, mature luminal cells and mature basal cells are maintained by lineage-restricted, unipotent progenitors, which are replenished by multipotent stem cells that are present during embryogenesis.

**Figure 4 cancers-11-01423-f004:**
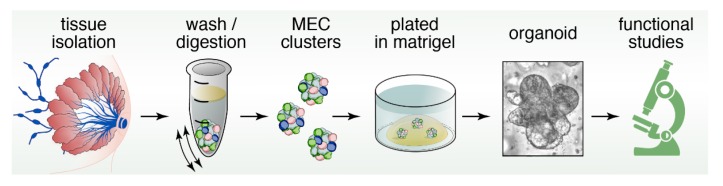
Pipeline of mammary gland organoid cultures. Submerging the Matrigel droplet with growth medium containing Wnt and R-Spondin ligands enables sustained maintenance of stem cell function and allows for functional studies on mammary stem cells (MaSCs) in these 3D cultures.

**Figure 5 cancers-11-01423-f005:**
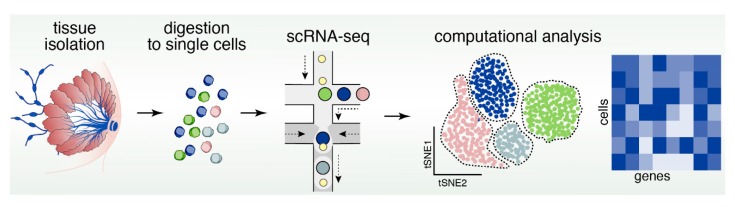
Pipeline of scRNAseq to resolve cellular heterogeneity with novel sequencing techniques. Processing mammary gland tissue of breast cancers into single-cell droplets coupled to single-cell RNA sequencing (scRNAseq) and data analysis (e.g., tSNE plots) allows for high resolution investigation of different cell types on the basis of individual cell transcriptome.

**Figure 6 cancers-11-01423-f006:**
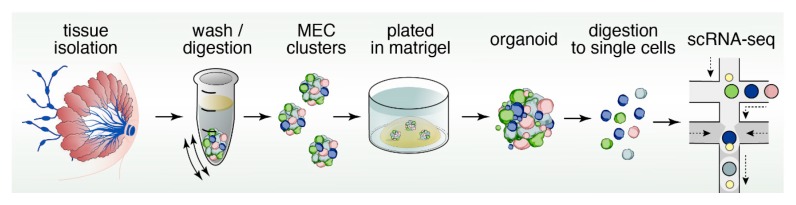
Organoids, scRNAseq, and other technologies to resolve cellular heterogeneity. Combination of different platforms such as organoids and scRNAseq but also mouse models, lineage tracing, and patient-centric assays will likely each make important contributions to resolve cellular heterogeneity in the mammary gland and in breast cancer.

**Table 1 cancers-11-01423-t001:** The EGFR family members and their interacting ligands. Adapted from Hynes and Watson [34].

EGFR Family	EGFR (ErbB1/Her1)	EGFR2 (ErbB2/Her2)	EGFR3 (ErbB3/Her3)	EGFR4 (ErbB4/Her4)
EGF	X			
TGF	X			
AREG	X			
EPG	X			
BTC	X			X
HB-EGF	X			X
EPR	X			X
NRG1			X	X
NRG2			X	X
NRG3				X

EGF: Epidermal Growth Factor; TGF: Transforming Growth Factor; AREG: Amphiregulin; EPG: Epigen; BTC: Betacellulin; HB-EGF: Heparin-binding EGF-like growth factor; EPR: Epiregulin; NRG: Neuregulin; EGFR: Epidermal Growth Factor Receptor; ErbB: Erythroblastic leukemia viral oncogene homolog; Her: Human epidermal growth factor receptor.

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
