# Peer review of "Unraveling Heterogeneity in Epithelial Cell Fates of the Mammary Gland and Breast Cancer"

_cancers, 2019, doi:10.3390/cancers11101423_

Round 1

Reviewer 1 Report

Comments to Manuscript Samocha et al.

This is a well written comprehensive review about the epithelial cell fates of the mammary gland and breast cancer. I really enjoyed reading it. However, I only have one minor comment:

1) Chapter 11: It is well-known that aberrant PI3K signaling is not only attributed to mutations in the PI3K gene, but also to overexpression of HER2 and HER3. HER2 overexpression is found in about 30% of breast cancer (the so-called HER2+ subtype) and is commonly associated with poor prognosis. Both receptors form the HER2/HER3 heterodimer, which has been determined as an oncogenic unit that drives breast cancer proliferation (Holbro et al., 2003 PNAS USA 100:8933). HER3 consists six docking sites for the p85 adaptor subunit of PI3K. Likewise, the interplay of HER2/HER3/PI3K and EGFR/HER2/PLCg1 has been shown to be important for breast cancer cell migration and dissemination (Balz et al., 2012 J Pathol 227:234.). In this regard, both EGFR homodimers and EGFR/HER2 heterodimers are potent inducers of ERK signaling, which would be in agreement with data of Huebner et al. (ref #84). So, it would be great if this topic could be discussed a little bit more in detail in this review.

Author Response

REVIEWER 1

This is a well written comprehensive review about the epithelial cell fates of the mammary gland and breast cancer. I really enjoyed reading it. However, I only have one minor comment:

1) Chapter 11: It is well-known that aberrant PI3K signaling is not only attributed to mutations in the PI3K gene, but also to overexpression of HER2 and HER3. HER2 overexpression is found in about 30% of breast cancer (the so-called HER2+ subtype) and is commonly associated with poor prognosis. Both receptors form the HER2/HER3 heterodimer, which has been determined as an oncogenic unit that drives breast cancer proliferation (Holbro et al., 2003 PNAS USA 100:8933). HER3 consists six docking sites for the p85 adaptor subunit of PI3K. Likewise, the interplay of HER2/HER3/PI3K and EGFR/HER2/PLCg1 has been shown to be important for breast cancer cell migration and dissemination (Balz et al., 2012 J Pathol 227:234.). In this regard, both EGFR homodimers and EGFR/HER2 heterodimers are potent inducers of ERK signaling, which would be in agreement with data of Huebner et al. (ref #84). So, it would be great if this topic could be discussed a little bit more in detail in this review.

We thank the reviewer for the excellent suggestions. We added a paragraph on EGFR signaling in cancer using the reviewer’s points. This is now part of chapter 11

Reviewer 2 Report

This was a well organized and written review article covering the heterogeneous nature of cell fate determination in the developing mammary gland.  I thoroughly enjoyed reading it. My only comments are that the last paragraph of section 7 seems a bit out of place with the following section on EGFR family and ligands. The authors might reconsider organization of those sections. Also while the figures are clear and well organized, it would be helpful to include a more detailed version describing the roles of the EGFR family and Wnt signaling in cell fate determination. There are some minor grammatical errors throughout. 

Author Response

REVIEWER 2

This was a well organized and written review article covering the heterogeneous nature of cell fate determination in the developing mammary gland.  I thoroughly enjoyed reading it. My only comments are that the last paragraph of section 7 seems a bit out of place with the following section on EGFR family and ligands. The authors might reconsider organization of those sections. Also while the figures are clear and well organized, it would be helpful to include a more detailed version describing the roles of the EGFR family and Wnt signaling in cell fate determination. There are some minor grammatical errors throughout. 

We thank the reviewer for the comments. We now restructured the review and reshuffled some of the chapters, which we believe flows better. We agree that we only touch on some aspects of Wnt and EGFR signaling (note that the Roose lab is a “signaling lab”). A comprehensive description of these signals, the cell fates, and heterogeneity would double the length of the review so we tried to be relatively brief here. We do agree with the reviewer that this is a very interesting topic and we are actively researching this. Perhaps a next review article.

Reviewer 3 Report

This review covered lineage trajectories of mammary epithelial cells, critical signal pathways for mammary development, and experimental techniques used to study stemness and cellular heterogeneity. This paper provided clear explanation about normal mammary development. However, the authors need to answer below several questions.

The authors introduced several transcription factors as specific markers for luminal and basal cells. I wonder if there are other lineage specific markers such as metabolism-related molecules.

The authors mentioned important factors to generate organoids. I have questions whether organoid culture from breast cancer tissues requires the same components, and whether there are studies applying organoid culture from breast cancer tissues to screen effective chemicals.

In terms of cellular heterogeneity in cancer, I recommend the authors to additionally introduce research articles which utilized single cell genomics for breast cancer study.

Author Response

REVIEWER 3

This review covered lineage trajectories of mammary epithelial cells, critical signal pathways for mammary development, and experimental techniques used to study stemness and cellular heterogeneity. This paper provided clear explanation about normal mammary development. However, the authors need to answer below several questions.

The authors introduced several transcription factors as specific markers for luminal and basal cells. I wonder if there are other lineage specific markers such as metabolism-related molecules.

In general, or most often, transcription factors are used as tools (markers) to track cell lineages. This is why we descried the transcription factors for luminal and basal cells, The ongoing single cell RNA seq efforts will undoubtedly lead to the discovery of novel markers. Possibly also metabolic enzymes.

The authors mentioned important factors to generate organoids. I have questions whether organoid culture from breast cancer tissues requires the same components, and whether there are studies applying organoid culture from breast cancer tissues to screen effective chemicals.

This is an active area of research. In fact we have two manuscripts, one mouse, one human, in preparation on these organoids and factors.

In terms of cellular heterogeneity in cancer, I recommend the authors to additionally introduce research articles which utilized single cell genomics for breast cancer study. 

We thank the reviewer for the excellent suggestions. We added a paragraph with references so that the reader can look these up and read this literature in depth.